# Access to Maternity Protection and Potential Implications for Breastfeeding Practices of Domestic Workers in the Western Cape of South Africa

**DOI:** 10.3390/ijerph20042796

**Published:** 2023-02-04

**Authors:** Catherine Pereira-Kotze, Mieke Faber, Luke Kannemeyer, Tanya Doherty

**Affiliations:** 1School of Public Health, Faculty of Community and Health Sciences, University of the Western Cape, Cape Town 7535, South Africa; 2Non-Communicable Diseases Research Unit, South African Medical Research Council, Cape Town 7505, South Africa; 3Department of Dietetics and Nutrition, University of the Western Cape, Cape Town 7535, South Africa; 4SweepSouth, Cape Town 8000, South Africa; 5Health Systems Research Unit, South African Medical Research Council, Cape Town 7505, South Africa

**Keywords:** comprehensive maternity protection, worker’s rights, breastfeeding practices, breastfeeding breaks, non-standard workers, domestic workers, mixed methods, South Africa

## Abstract

Access to comprehensive maternity protection could contribute to improved breastfeeding practices for working women. Domestic workers are a vulnerable group. This study aimed to explore perceptions of and accessibility to maternity protection among domestic workers in the Western Cape, South Africa, and potential implications of maternity protection access for breastfeeding practices. This was a mixed-method cross-sectional study including a quantitative online survey with 4635 South African domestic workers and 13 individual in-depth interviews with domestic workers. Results from the online survey showed that domestic workers had inconsistent knowledge of maternity-protection entitlements. Data from individual in-depth interviews showed that most participants struggled to access all components of comprehensive maternity protection, with some entitlements being inconsistently and informally available. Most domestic workers were unfamiliar with the concept of breaks to breastfeed or express milk. Participants provided suggestions for improving domestic workers’ access to maternity protection. We conclude that improved access to all components of maternity protection would result in improved quality of care for women during pregnancy, around the time of childbirth and on return to work, and for their newborns, especially if an enabling environment for breastfeeding were created. Universal comprehensive maternity protection could contribute to improved care for all working women and their children.

## 1. Introduction

Comprehensive maternity protection incorporates health protection at the workplace, a period of maternity leave, cash and medical benefits while on maternity leave, non-discrimination and job security for pregnant and breastfeeding women, breastfeeding breaks on return to work, and support to access childcare [1]. The International Labour Organization (ILO) Maternity Protection Convention No. 183 of 2000 (article 10) states that women have the right to one or more daily paid breastfeeding breaks [2]. The ILO Maternity Protection Recommendation No. 191 of 2000 further recommends that facilities for breastfeeding under hygienic conditions should be available at or near the workplace [3]. The workplace and employment are key settings where adequate legislation, policy, financing, and monitoring and enforcement of policy and legislation could contribute to an enabling environment for breastfeeding [4]. Availability of and access to all components of comprehensive maternity protection could therefore contribute to improved breastfeeding practices. In South Africa (SA), only 32% of infants under six months are exclusively breastfed [5], but the World Health Assembly has recommended the target of at least 50% for all countries by 2025 [6] and at least 70% of infants to be exclusive breastfed by 2030 [7].

Domestic workers can be considered a category of informally employed wage workers [8] and often cannot access legal and social protection [9]. Non-standard employment refers to temporary, part-time, multi-party, or disguised employment or dependent self-employment [10]. Non-standard workers do not have permanent, full-time positions usually associated with a set of employment benefits (such as paid leave, pension contributions, sick leave, etc.) Domestic workers are a vulnerable group of non-standard workers, with recent research documenting various human-rights violations of live-in domestic workers in SA [11]. Globally, domestic workers struggle to access social-security rights, with only one in five domestic workers covered by social insurance schemes [9]. Domestic work continues to be undervalued, under-recognised, and managed informally. The ILO suggests that social protection has the potential to enable domestic workers to transition to more formal employment, and that the aim should be for domestic workers to be treated as favourably as other workers [9].

Platform work (also referred to as “digital labour platforms” or “the gig economy”) is one form of non-standard employment that involves an employment intermediary connecting employers through the internet and is enabled by online and digital mechanisms for independent contractors. In this way, the platform acts as an employment broker [12]. Platform work has been described as creating new income-generation opportunities, making certain services more accessible and allowing flexibility and autonomy in work [12]. In the domestic-work sector, technology has been described as potentially improving employment conditions by standardising wages and professionalising domestic work [13]. In SA, domestic workers reported joining a platform to seek employment opportunities, including possible permanent work, due to distrust of other recruitment agencies and for potentially more consistent and higher earnings [13]. However, there are some disadvantages to platform work, such as intense competition between workers, leading to low wages, potentially inferior working conditions, and economic insecurity. In SA, domestic workers reported irregularity of work, transport, and data costs as disadvantages to platform work [13]. The nature of the employment relationship between platforms and workers also remains unclear. Different countries have started to grant different levels of labour protection and regulation to platform workers [12].

In SA, the maternity-protection policy landscape has recently been described, including for domestic workers, and has been found to be fragmented and difficult for employers to determine their responsibilities and for workers to know their entitlements [14,15]. A comprehensive description of maternity protection in SA has been provided elsewhere [14], but a summary is available in Table 1. Although most provisions of maternity protection apply to all working women, including domestic workers, it is more difficult for non-standard workers like domestic workers to access all components of maternity protection [15].

Maternity protection is one component of social protection, but especially important in a sector like domestic work where the majority of workers are women [16]. Domestic workers make up a large proportion of the female workforce—almost 76 million workers globally and 11.9% of the female workforce in SA [16,17]. It is therefore important to consider their access to labour-related protection, including maternity protection. Previous research has mainly focused on individual components of maternity protection, such as maternity leave, breastfeeding breaks, or childcare, and the possible implications for breastfeeding practices [18,19,20,21]. It is important to explore the implications of access to all components of comprehensive maternity protection for domestic workers to support this cadre of workers to reach their breastfeeding goals and because of the benefits to infant and young-child health and development that would result from ensuring recommended breastfeeding practices [22]. Therefore, this study aimed to explore perceptions of and accessibility to maternity protection among domestic workers in the Western Cape and South Africa, and the potential implications of maternity-protection access for breastfeeding practices.

## 2. Materials and Methods

This was a mixed-method cross-sectional study that included data from a quantitative online survey with 4635 South African domestic workers and 13 individual in-depth interviews (IDIs) with domestic workers.

### 2.1. Study Setting

The online survey was distributed to domestic workers across SA. SA is a middle-income country with high rates of poverty, inequality, and unemployment [17]. Female labour-force participation in SA in 2022 was 53% [23]. In 2022, 69% of working women were employed in the formal sector, 14.7% in the informal (non-agricultural) sector, 12.4% worked in private households, and 4% worked in agriculture. Of all working women, 11.9% were domestic workers [17].

The IDIs were conducted in the Western Cape, the third most populous province in South Africa, accounting for 7,212,142 (11.9%) of the population [24] and with approximately four million people living in the Cape Town metropolitan area [25]. There are approximately 113,000 people working in private households in the Western Cape [17].

### 2.2. Participant Sampling and Selection

For the online survey, a private company that manages an online matching platform for domestic workers (through a home-cleaning-service application that connects workers with once-off and recurring clients) distributed a link to the online survey to all domestic workers currently active on their platform, domestic workers who were previously active on the platform, and all those who had made applications to join the platform, and encouraged recipients in the message to forward it to other domestic workers. The participant sampling and selection process is illustrated in Figure 1. From this distribution strategy, 4635 domestic workers working in SA completed the full online survey. For this study we were given access to responses from the 2625 domestic workers, who were the participants that consented to answering three questions about maternity protection (see Appendix A). The socio-demographic characteristics of these 2625 domestic workers are presented in Table 2. The results from this are presented in Section 3.1.

The results from the three questions about maternity protection answered by 2625 domestic workers are presented in Section 3.1 Domestic worker perceptions of maternity protection entitlements (including Table 1 and Table 2). The results from the individual in-depth interviews with domestic workers working in the Western Cape are presented in Section 3.2 Domestic worker access to comprehensive maternity protection. 

For the IDIs, participants were recruited from amongst those who had completed the online survey. Of the 4635 South African respondents that completed the full online survey, 2717 indicated interest in participating in a follow-up interview to answer questions about maternity protection. The inclusion criteria used were that participants needed to be female, have delivered a baby in the past three years, be living in the Western Cape, be South African, be able to communicate in English, be between the ages of 18 and 49, and have provided a mobile phone number. After applying the eligibility criteria, 194 were contacted for interviews. We sent a WhatsApp or text message followed up with a phone call where necessary to these respondents asking whether they were still interested in participation, confirming eligibility for participation, and determining availability and willingness to take part in an in-person interview. Of the 194 contacted, 181 were not interviewed either because they did not meet the eligibility criteria or they were not available or willing to take part in an individual in-person interview. Following this screening process, we completed interviews with 13 domestic workers, and these results are presented in Section 3.2.

### 2.3. Data Collection

For the online survey, a link was sent to participants via WhatsApp and SMS (short message service, or text message) and they were able to complete the survey data-free (i.e., without needing to pay for internet access) on their smartphone or a computer. The survey was administered through Survey Monkey. Three questions on perceptions of access to maternity-protection entitlements were included in the questionnaire (Appendix A).

For the IDIs, a semi-structured interview guide was used to guide the IDIs (see Appendix A). The IDIs were conducted in person in February 2022 and all necessary COVID-19 protocols were observed. All interviews were conducted in English as all the domestic workers could understand and respond in English. Interviews ranged between 24 and 42 min but were on average 35 min long. All interviews were conducted by the first author (CPK) and then transcribed.

### 2.4. Data Analysis

For the online survey, data were exported from SurveyMonkey to Microsoft Excel and basic descriptive frequencies were calculated in Microsoft Excel.

One researcher (CPK) manually analysed the IDIs using a thematic-analysis approach [27,28]. Analysis began with familiarisation of the transcript contents by reading and checking the transcripts. Using a deductive approach, information was coded according to the components of comprehensive maternity protection: health protection, maternity leave, cash and medical benefits, employment protection, non-discrimination, breastfeeding breaks, and childcare support. As the authors became familiar with the interview content, sub-themes were developed and refined and then linked to the main themes (Table 3). A reflexivity journal was kept throughout the analysis process documenting any personal characteristics of the researcher that may have influenced their analyses.

### 2.5. Ethics Approval

All participants that completed the online survey provided consent electronically. For the individual IDIs, all participants provided written informed consent and agreed to interviews being audio-recorded. Interview data were stored electronically and securely by the first author (CPK). Participants’ confidentiality was maintained by removing names and personal information linked to individuals’ responses from the transcribed data in the reporting of the results. Privacy, confidentiality, and anonymity were ensured. Ethical approval was obtained from the University of the Western Cape’s Senate Research Committee and Ethics Committee (Reference Number: BM20/5/7).

## 3. Results

### 3.1. Domestic Workers’ Perceptions of Maternity-Protection Entitlements

The socio-demographic characteristics of the 2625 domestic workers who answered the questions on perception of maternity-protection entitlements are presented in Table 2. Most respondents (96.7%) were female, and 60.7% were foreign nationals (i.e., not South African citizens). Most of the sample (55.4%) worked in Gauteng province. Two-thirds of the sample (64%) had either two or three children. Just over half (51.9%) earned between ZAR 2000–4000 (USD 112–223 on 7 November 2022) [26] per month from domestic work. Only 7.7% reported contributing to the national social insurance scheme, called the Unemployment Insurance Fund (UIF), which protects employees in the event of unemployment and covers those in the formal economy and domestic workers and farmworkers.

Of the 2625 domestic workers, around two-thirds believed that they should be allowed to receive paid time off to attend antenatal clinic visits. Only half thought that they were entitled to job security because of a pregnancy. Less than half believed that work-related duties should be amended to accommodate pregnancy or that they should not be discriminated against because of a pregnancy (Table 3). When asked which maternity benefits they thought they should be entitled to receive, 26.2% thought they should have four months of full paid maternity leave organised by the employer, 21.5% felt they should get four months of maternity leave and claim from the Unemployment Insurance Fund (UIF), 20.8% thought they should get some (between 6 weeks and 4 months) maternity leave, and 20% thought they should receive four months of partially paid maternity leave (Table 3). Only 5.8% of respondents thought they should receive either no maternity leave or four months of unpaid maternity leave, and 5.8% did not know. When asked about other components of maternity protection, 60.2% of domestic workers felt they should be able to have paid time off work to attend postnatal clinic visits, 15.7% felt they should be able to have unpaid leave to attend postnatal clinic visits, 18.3% thought they should be entitled to daily breastfeeding or expressing breaks, and 6.7% thought they should be able to bring their baby to work with them. A total of 10.6% of respondents felt they would not be able to access any additional maternity protection benefits and 9.3% did not know which other maternity benefits they should be able to access.

### 3.2. Domestic Workers’ Access to the Different Components of Maternity Protection

#### 3.2.1. Socio-demographic Characteristics of Domestic Workers Included in the Sample for IDIs

All 13 domestic workers interviewed were Black African women living in Western Cape province who had delivered a baby in the past three years relative to the interview date. Six mothers had delivered their babies in the past year, so their child was younger than 12 months; four women had a child between 13 and 24 months of age; and three had a child between 25 and 36 months of age. Participants were between 26 and 42 years of age. Participants had between two and five total other children, with the youngest child being under the age of three years. Four participants reported formula feeding from birth, six were still breastfeeding their child, and three had started breastfeeding but were either mixed feeding or had changed to formula when they started working. Six participants accessed work through an online platform (referred to in the results as “platform workers”), four through private clients, and three through a combination of both. Themes were developed and grouped according to the six components of comprehensive maternity protection (Table 4).

#### 3.2.2. Health Protection at the Workplace for Domestic Workers Is Employer-Dependent

Participants described how access to health protection while pregnant is unpredictable and dependent on the individual employer. Some employers appeared to be understanding and allowed certain reductions in the workload expected of domestic workers due to pregnancy:


*“Washing the windows. Clean the windows. I couldn’t climb up. Or to clean all the cupboards. Or carry some heavy stuff. I couldn’t. They were very, like understanding. I didn’t do all of those things. I think they understand, because I was pregnant. I didn’t even have to tell them, that I can’t do this, so sometimes they will just know. Just do this, just do the basics, then you go, because maybe you are tired. So, I think they were very supportive.”*
(Domestic worker, DW2, worker for private clients)


*“Like moving the things. The hard things, like the fridges and stuff. I, I told her that I’m no longer going to, but she’s the one, she was straight. She’s the one who told me, you must not move anything now that you’re pregnant, because it’s gonna hurt your baby. So, whenever that needs to be moved, she asked someone else to do that.”*
(DW8, worker for a private client)

Other employers expect domestic workers to carry on as normal when pregnant or soon after and while still recovering from childbirth:


*“Sometimes, if you work here in [two Cape Town suburbs], yoh! People from there. No, they, even if they see that you’re, like you’re struggling, they want to scrub to, kneel down. So, it’s very difficult.”*
(DW11, platform worker)


*“Like the one I was last, I’m a Caesarean person. Like, I have to take it with some small break. To go there, in there, and put my bandage because my operation wasn’t, not yet healed… It was sore. I still had to work.”*
(DW12, platform worker)

One participant described how she worked until her due date for delivery and that her employer had to take her to hospital to give birth:


*“It was hard, hard to work while I’m pregnant because I told them, I asked, maternity leave. They said, like the customer, the client, I asked to take leave. They said no. You can’t because we’re gonna need someone here. I have to work till my due date. Then I woke up at one in the morning on my due date. I called the bosses. They came down. They took me to hospital. Then I left for hospital, when I’m done everything, I gave birth. I go. I went home. I didn’t get paid.”*
(DW12, platform worker)

##### Access to Health Care for Domestic Workers during the Antenatal and Postnatal Periods Necessitates Unpaid Leave

In SA, healthcare is available free of charge to women and provided through public health services for antenatal, childbirth, and postnatal care. Most participants described how, if they needed to attend the health clinic for routine appointments during pregnancy (antenatally) or soon after birth (postnatally), they would plan to attend these appointments on days when they were not working. This time taken to attend health visits is therefore unpaid leave. Some participants described that employers would allow flexibility but if domestic workers did attend a clinic on a workday, they were not paid, or the payment was decreased:


*“If I will go to the clinic, I must take a day off.”*
(DW1, worker for a private client)


*“No, she didn’t say anything about that. But she told me, she told me if I need to go see the doctor, I must tell her, if maybe, that day that I should come to work, is the same day that I should go to the clinic, I just have to tell her, then we’ll redo the schedule.”*
(DW8, worker for a private client)


*“Yes, I was able but if I go to the clinic the money is cut. Ja, because at the clinic, er, it’s busy in our clinic, so we spend almost a day, almost the whole day. Ja, so there’s no use to go to work maybe around three or two.”*
(DW13, worker for private clients and platform worker)

#### 3.2.3. Some Domestic Workers Experience Discrimination due to Pregnancy and Childbirth

Although many domestic workers described how their employers’ response to their pregnancy was that they were happy for them and excited to find out they were pregnant, there were some participants who experienced discrimination due to pregnancy. A number of participants described how the platform or agency deactivates workers when they are pregnant and go on maternity leave: 

*“They say there at work* [online platform], *if you’re pregnant, they’re going to deactivate you. Then you must tell them, when you come back, they’re gonna interview again.”*(DW3, platform worker)

“Temporary deactivation” applies to any length of time more than 30 consecutive days where a worker chooses to deactivate themselves for discretionary reasons (including maternity leave). Deactivation means that a domestic worker will not receive new work from the platform during this time. The domestic worker does not need to reinterview unless they are away from the platform more than 6 months.

One participant described how upon hearing the domestic worker was pregnant, her employer suggested that she terminate the pregnancy:

*“No, afterwards she said, I must abort the baby. I said no. Because she knows me, a long time ago, I got three children already. The first born was one. Then I got a second child. They are twenty-one now. The oldest one is twenty-five now. The second ones are twins. So, she said to me, how could you have another baby again? You know? So, I said, no, I’m not gonna kill my baby. I’m gonna have my baby, and then I’m gonna close it* [perform a sterilisation procedure]. *It’s a mistake. I know it’s a mistake, but I’m not gonna abort.”*(DW9, worker for a private client)

Another participant described how her employer changed her attitude towards her when finding out she was pregnant, and then was told not to return to work at all from when she was 6 months pregnant: 


*“So after she found out, I saw the changes in her like she’s no longer the same, I don’t know. Not at all alright. I’m not supposed to be pregnant while I’m working.”*
(DW13, worker for private clients and platform worker)

#### 3.2.4. Many Domestic Workers Experience Job Insecurity Due to Pregnancy and Childbirth

Most participants reported that when they found out they were pregnant, they were concerned about job insecurity and did not feel they were guaranteed employment protection:


*“But to have a baby, you must say you’re gonna lose this job. It’s not easy to have a baby there at online platform.”*
(DW3, platform worker)

Some participants lost jobs [i.e., clients] due to pregnancy: 


*“Because, some I did lose because there are people who say, no, she’s pregnant. Now we can’t work with her. So, I lose some, four jobs that I lose. Then I started the new ones.”*
(DW4, platform worker)

Some participants were worried they would not have a job after maternity leave because their employer would have found someone else to work while they were on leave:

*“If I was on maternity leave for three months, maybe she* [the employer] *will book someone else and then she will say: ah, this one is good. More than me. And then they will hire them.”*(DW1, worker for a private client)


*“Yes, when I was pregnant, I was worried. I thought they’re gonna put another one. They’re gonna replace someone in my place.”*
(DW12, platform worker)

One participant described losing her job due to pregnancy and was surprised by this:


*“Because when I was still pregnant… I felt like I can’t handle the job. I said to her—no ma’m, I need to rest, because I feel the pain if I work hard. Then I need to rest and then I will come back when I deliver. And then she said fine. We didn’t fight. We didn’t do anything. She said it’s fine. I say to her, can I bring someone to step in for me while I’m in maternity? She said, no, no, no, no. I will wait for you. The minute you feel okay, you can just phone me and then you come back. So we didn’t fight. We didn’t do anything… I feel like that lady betrayed me. She was supposed to pay me. Ja. If she fires me, she’s supposed to pay me. And she knows that thing’s wrong”*
(DW9, worker for a private client)

#### 3.2.5. Difficulties for Domestic Workers to Access Paid Maternity Leave

##### Domestic Workers Are Unable to Access Cash Payments While on Maternity Leave despite Legal Eligibility to Social Insurance in SA

In the online survey (*N* = 2625), only 7.7% of domestic workers (*n* = 203) indicated that they were registered for the UIF, with the remainder (92.3%; *n* = 2625) indicating that they were either unsure of UIF registration or not registered for the UIF. In the individual IDIs with 13 domestic workers, all participants stated that their employers were not contributing to the UIF on their behalf and therefore could not access cash maternity benefits eligible to them:

*“They* [online platform] *don’t want us to have a UIF. I don’t know, they don’t explain to us why they don’t have a UIF. We don’t have leave, if you’re on leave, you’re unpaid. I don’t know why they do this.”*(DW4, platform worker)


*“They are not deducting, the UIF money. You see? Now I’m stranded, I’m having a baby, there was no maternity leave benefit. No provident fund, no nothing. If you don’t work, you don’t get paid. So, I’m gonna find another job or else I will go back to security…. I will go back to my security job, where there are, benefits, like UIF and provident fund.”*
(DW1, worker for a private client)

A few participant responses indicated limited knowledge of eligibility to access the social insurance scheme in SA (the UIF). Many participants appeared uncertain of the specific benefits of contributing to the UIF:


*“I heard about the UIF, but I don’t know.”*
(DW12, platform worker)


*“I’m not sure. How does it go? How do you get registered and stuff like this, so I’m not, educated on how to do that…. I don’t know how much are they going to take it from my money to, to pay the UIF? Or they are going to contribute? I am not sure. I am not totally sure about that.”*
(DW2, worker for private clients)

A common theme was that participants had heard of the UIF but were uncertain of the details. Some participants described that the UIF could provide some income replacement if they were retrenched or not working due to maternity leave:


*“I think sometimes you lose your job… sometimes you are pregnant and then you have to receive some money to feed your baby.”*
(DW11, platform worker)


*“It’s because when the time that, when I’m sitting down, like the time I was on maternity leave, I was supposed to get money. But I didn’t get money because I wasn’t working. So, I think it’s very important to get UIF. So that when you’ve got a problem, you can claim UIF, if you don’t have the money.”*
(DW8, worker for a private client)

Because participants were recruited through an agency or platform, some described that they expected the platform to take responsibility for ensuring access to social insurance: 


*“And they always answer us and say, we are not employers. We are the platform. So, we just keep quiet. We don’t know where to go. And we are scared to be fired, while we still need a job.”*
(DW11, platform worker)

##### Unpaid Maternity Leave Is the Only Leave Option for Domestic Workers but Is Unaffordable and Therefore Inaccessible

Almost all participants described that any period of maternity leave taken was unpaid: 

*“No, they* [manager at online platform] *just say I must take, if you have a maternity leave, no work, no pay.”*(DW3, platform worker)

The shortest length of maternity leave reported was two weeks: 


*“I did take off… fourteen days. I didn’t get paid. Because there’s no food on the table, so I got up and go and work.”*
(DW3, platform worker)

The duration or maternity leave reported varied from two weeks to six months, with most participants reporting taking around three months of unpaid leave around the time of childbirth.

One participant described not taking maternity leave: 


*“I didn’t take maternity leave… Because they don’t have anything to contribute to me. So, I must go to work.”*
(DW4, platform worker)

One participant was unsure as to whether she was entitled to maternity leave: 


*“I don’t think, I think we have the maternity leave, but I don’t think so. They deactivate you on the platform. That’s just only thing I know. You don’t get nothing.”*
(DW10, platform worker)

When asked whether they would want longer or paid maternity leave, many participants indicated that they would rather work to earn money or have access to social insurance (the UIF in SA): 


*“I don’t mind for maternity leave, but for the UIF. I think it, every worker, it’s necessary to have UIF.”*
(DW4, platform worker)

##### Inaccessibility to Social Insurance Results in Dependence on Social Assistance, Which Also Has Challenges

Since all participants indicated that they were unable to access social insurance, when asked how they managed financially during the period of unpaid maternity leave, some participants mentioned receiving social assistance. Therefore, during the interviews, we sought to explore more about social assistance. From the responses received, it seems like social assistance, in the form of welfare grants, is more accessible than social insurance in SA, even though there are still some challenges experienced, such as time-consuming and costly processes required for applications:

*“You must wait for birth certificate, clinic card, and then there by the hospital, they don’t send the social workers to do grants for you. They just send someone from home affairs, to do the certificate, which is right. But for SASSA* [South African Social Security Agency, responsible for administration of the Child Support Grant], *there’s not someone there at the hospital asking do you need a form to apply for the grant for SASSA. You must wake up early in the morning. Four o’ clock you must be out of your house... Five o’ clock you take queue. There are many people there. So maybe you’re gonna sit outside with this small child… If your child is hungry, they’re gonna attend you at one o’ clock. It’s the first day you apply. Then you must wake up again. Four o’ clock to get this child grant again. They can attend you one o’ clock again. To bring back the forms. It takes a long time and there in [an informal settlement], we don’t have SASSA. We must take a taxi to Cape Town. It’s very difficult for us as a domestic worker.”*(DW3, platform worker)

Two participants described alternative strategies they used to access the Child Support Grant due to challenges experienced. One participant travelled to a different province (the Eastern Cape, approximately 900 km away from Cape Town) to request her mother-in-law to apply for the grant and then transfer the money to her, because this was felt to be quicker than applying for the grant from her own city.


*“So, I have to ask my mother-in-law to make the grant there by Eastern Cape because it’s very easy to make the grant in Eastern Cape… She transfers the money to me. I go there to give my, mother the certificate and the card and I came back. And my mother did everything there… Yoh, a lot of times and I get the date there by [suburb in the Western Cape] and I was unlucky that day because there was a noise and they said; no, you must go to [different area in the Western Cape] and other areas… So, I was in need. So that’s why I asked my mother to.”*
(DW11, platform worker)

When probed, this participant shared that this cost her ZAR 1200 (approximately USD 67) for a return journey to the Eastern Cape and that she took her newborn baby with her on the trip. The monthly value of the Child Support Grant for all women is only ZAR 480 (approximately USD 27) [29].

#### 3.2.6. Domestic Workers Are Not Familiar with the Entitlement to Breastfeeding Breaks upon Return to Work

All participants were unfamiliar with the concept of breaks to breastfeed or express as a component of maternity protection. Even those who had other children had not heard of breastfeeding breaks before: 


*“It’s the first time I hear about it.”*
(DW2, worker for private clients)

One participant who was leaving expressed breastmilk for her child when she went to work described how she expressed or breastfed when at home but had not ever expressed breastmilk at work: 


*“I only express when I’m at home”*
(DW5, worker for private clients and platform worker)

One participant questioned how a domestic worker would be able to have a breastfeeding break, because her child would probably not be near her workplace: 


*“How do they do that? Because maybe I’m working in town and my baby is at the crèche, maybe with my sister at my house. So, if I had a break, I have to go home. Or what?”*
(DW8, worker for a private client)

Two participants responded that knowledge about breastfeeding or expressing breaks would help and described that not everyone knows about these breaks:


*“To get the knowledge about the breaks, yes, and breaks to express the milk, yes. I think most people need to know about that. Like because I didn’t know either.”*
(DW13, worker for private clients and platform worker)


*“I think the, what you’re talking now about, that’s of the breastmilk breaks? I think that if people knew about it, I think they will just take their break and try and do the, express. And if they know, if they are guaranteed that their jobs are not at stake, maybe.”*
(DW2, worker for private clients)

Although it was not formally classified as a breastfeeding break, one participant described how her employer encouraged her to bring her baby to work and breastfeed the child during the workday: 

*“She’s* [the employer] *the one who said, I must come to work with him. She gives me time to breastfeed him.”*(DW1, worker for a private client)

#### 3.2.7. Challenges with Storing Expressed Breastmilk at Work

Participants described several challenges with storing breastmilk that they had expressed during the day at work, with some participants indicating that others may have negative perceptions of expressed breastmilk when asked whether they would be able to put the milk in a fridge at work:


*“Like the breastmilk is not, like when it’s in a bottle, it’s not, like a nice colour, you know? They can’t put it in the bag too because during the day, the milk is gonna go sour.”*
(DW9, worker for a private client)


*“I think it will depend on the belief of that someone. Because some, they don’t have the information. Some, they don’t like even the breastmilk… Even my husband, like the first time I put it in the fridge, they laughed and; please don’t use the, like the container. I’m going to use it again.”*
(DW11, platform worker)

One participant indicated that she could probably store the milk in a fridge in the garage: 


*“There’s two fridges in the garage, where they put some things. So, I, if I go to her again, I’ll talk to her and then I’ll leave it [expressed breastmilk] there. I don’t think she can refuse.”*
(DW6, worker for private clients and platform worker)

#### 3.2.8. Domestic Workers Struggle to Access Childcare on Return to Work

Domestic workers employed through the platform or agency reported that they were told not to take their child to work with them: 


*“But in the online platform they tell us in the booking, you are not allowed for the baby. If you grab your booking, don’t go to the client’s house with the baby.”*
(DW3, platform worker)

Many participants spoke about the high costs of childcare and that sometimes up to half the money earned from domestic work can go to childcare and transport to get to work:

*“There’s nothing, which I’m benefitting. It’s just a loss… You’re just working only for the transport…. Because when you count that money, a day I work for one twenty or one forty* [ZAR = USD 7 or 8]. *It can’t reach to where I want it to go, but at least, I don’t sleep hungrier, at least, but it can’t take me anywhere…. Like the hours? The money is small.”*(DW7, platform worker)

One participant described that the crèches (a nursery or day care centre where infants and young children are cared for during the working day) close to where a domestic worker lives are more affordable than crèches close to where they work: 

*“So the crèches in town, in our location is, the money. It’s cheaper in the location. But we have to leave him the whole day in the location* [i.e., township or informal settlement].*”*(DW13, worker for private clients and platform worker)

#### 3.2.9. Domestic Workers Provided Suggestions for Improving Access to Maternity Protection

One participant suggested that the Department of Employment and Labour should do more inspections at households as workplaces:

*“I did hear that there’s some people that get in the houses and ask for the one who have a domestic worker. They ask—but, even one day I didn’t see them at my work. Because I was thinking maybe why they don’t, why they* [Department of Employment and Labour] *don’t come here and ask for it, so that I can be registered.”*(DW6, worker for private clients and platform worker)

One participant felt that it would help with childcare responsibilities if domestic workers could bring their children to work with them:


*“If the employers, they can say; you are welcome with your, child. You can take the child with you if you’re going to work. Because we do the house chores with the babies. We can put the baby on our backs and still work, even at home. So, we can do that at the employer’s house. So, they can take the babies with them if they don’t have the money to take them to the crèche or they are still too young to go to the crèche”*
(DW8, worker for a private client)

However, another participant disagreed with this, saying: 


*“I don’t think you can work nicely when the baby is around.”*
(DW9, worker for a private client)

## 4. Discussion

The results from the online survey with domestic workers in SA reveal inconsistent knowledge of maternity-protection entitlements. From the results of the individual IDIs conducted with domestic workers in the Western Cape, most participants struggled to access all components of maternity protection. From our results, it seems that the inaccessibility of maternity protection for domestic workers is due to a variety of reasons, including limited knowledge and awareness of both domestic workers and their employers regarding entitlements, as well as systemic problems with the implementation of these provisions’ entitlements. Health protection at the workplace, access to medical benefits, and maternity leave appeared to be conditionally dependent, with health protection being informally and inconsistently available. Many domestic workers reported using unpaid time off work to attend health-facility visits, and maternity leave was available but unaffordable due to the inability to access cash payments while on maternity leave. Several participants reported experiencing discrimination due to pregnancy and job insecurity due to pregnancy and childbirth, and most domestic workers were unfamiliar with the concept of breaks to breastfeed or express milk and many struggled to access childcare upon return to work. As described above, participants provided some suggestions for improving domestic workers’ access to maternity protection.

### 4.1. Domestic Workers in SA Are Unaware of Their Maternity Protection Entitlements

The results from the online survey demonstrating inconsistent knowledge of maternity protection revealed especially wide variation regarding perceptions of cash payments on maternity leave. This is plausible given the fragmented maternity-protection policy environment in SA [14]. Less than one-fifth of respondents from the online survey thought they should be entitled to daily breastfeeding or expressing breaks. From the individual interviews with 13 domestic workers, none were familiar with the concept of breaks to breastfeed or express as a component of maternity protection. Although research has been conducted on the availability of breastfeeding breaks in national policy [30], there is no research available on employee or worker knowledge of breastfeeding breaks. Only two studies have been conducted (both in the USA) on knowledge of breastfeeding laws. One small American study described that just under half (47.8%) of employers had heard of the Nursing Mothers law [31]. A study conducted in the USA to determine awareness of breastfeeding laws among students and staff at institutions of higher learning showed that although awareness of breastfeeding laws and provisions among respondents was low, just over half reported that their institution provides a supportive environment for breastfeeding [32]. A recent study conducted in Vietnam with formally employed women revealed high awareness and uptake of the advanced maternity-protection policies but also many implementation gaps and lack of knowledge of the full set of maternity entitlements provided by law [33]. Research on employee and worker awareness of maternity-protection legislation is needed, especially amongst non-standard workers.

In SA, a civil-society organisation has developed a general guide on domestic-worker rights aimed at employers [34]. It would be helpful to have a user-friendly guide in all 11 official SA languages (English, Afrikaans, Zulu, Xhosa, Sepedi, Tswana, Southern Sotho, Tsonga, Swazi, Venda, and Southern Ndebele) summarising the maternity-protection entitlements available for domestic workers to be distributed to both domestic workers and their employers. Furthermore, the implementation of social- and behavioural-change communication campaigns, including the use of digital technologies, should be considered to improve the availability of up-to-date information on worker rights and employer responsibilities. In SA the National Department of Health implemented a mobile health programme called MomConnect, which aims to improve maternal health through the provision of targeted health-promotion messages sent via text to mobile phones of registered users [35]. A similar programme could be considered to share updated information on labour entitlements, including those for maternity protection, for which pregnant workers and employers could register.

### 4.2. Some Components of Maternity Protection Are Available to Domestic Workers in SA, but Are Inaccessible

The results from the individual IDIs in this study show that domestic workers are unable to access most components of maternity protection throughout the perinatal period, from pregnancy to the first years of the child’s life. This is concerning since most domestic workers are women and often of childbearing age, and therefore need to combine income-generating activities with their own unpaid care work and reproductive responsibility. The implications of inaccessibility to maternity protection are that many women return to work early and therefore may not have adequately recovered from childbirth, bonded with their newborn baby, and established breastfeeding [36]. Limited research has been done on access to maternity protection for domestic workers. An ILO policy brief on maternity protection and work–family measures for domestic workers reporting on data from 2011 showed that globally, 35.9% of domestic workers had no legal entitlement to maternity leave, with higher rates of domestic workers not accessing maternity leave reported in Asia and the Pacific [37]. This report indicates that globally, 62.7% of domestic workers have legal entitlements to maternity leave, but this does not guarantee practical access, and the report acknowledges that restrictive prerequisites and eligibility criteria can restrict access for domestic workers. Globally, 39.6% of domestic workers do not have legal entitlement to maternity cash benefits [38].

In SA, domestic workers have had legal access to maternity cash benefits since 2002 through a sectoral determination for domestic work. However, this only applies to those working more than 24 h per month per client or employer, which is an obstacle to many domestic workers who work for different employers for shorter durations per month. Furthermore, recent stakeholder engagement revealed implementation challenges resulting in limited access to cash payments for domestic workers while on maternity leave [39]. Therefore, policy implementation is hindered by the constraints and the occupational reality of domestic workers. It would therefore be beneficial for the government to shift away from top-down policy development and to involve key stakeholders from the domestic-worker sector when legislation is developed and when policy implementation is considered. There has been some engagement in SA where relevant stakeholders have come together to discuss domestic-worker labour issues, but this has mostly been driven by civil society [40].

#### 4.2.1. Challenges in the Implementation of Social Insurance in SA

Social protection schemes can include various combinations of social insurance (where employers and employees usually contribute a percentage of monthly wages to a government-managed fund, from which eligible beneficiaries can apply) and social assistance (non-contributory, tax-funded benefits usually in the form of cash transfers) [41]. There are examples of successful extensions of social protection coverage to workers in the informal economy through both routes, as well as the facilitation of the transition of informal workers to the formal economy [41]. Both quantitative and qualitative data from our study demonstrate low levels of access to the national social insurance scheme in SA (provided through the NDEL), and participants resorting to social assistance (provided through the National Department of Social Development, NDSD) in the absence of adequate cash benefits while on maternity leave. In the Philippines, many women in the informal sector are also not members of the social insurance programme [42]. One consequence of this use of social assistance instead of the maternity benefit from social insurance is that for some women, the economic value of social assistance available in SA may be less than the amount of income replacement a woman would receive through social insurance. Since access to social assistance seems to be better than access to the national social insurance scheme, it could be helpful for government departments (for example, NDEL) to learn from each other (e.g., NDSD) to improve access to services. Recent research conducted with workers on the platform used in our study revealed complications and unclear messages from platform management related to how platform workers could access social insurance [13]. A framework for platform workers to access the national social insurance scheme may be needed. However, the challenges that domestic workers in SA have experienced in accessing social insurance over the past 20 years suggest that social assistance strategies may be more effective.

#### 4.2.2. Limitations to the Enforcement of Maternity-Protection Legislation in SA

To improve domestic workers’ access to maternity protection, one IDI participant in our study suggested improved inspections by the NDEL at households (the location of workplaces for domestic workers). Inspections at households may be complex to organize, due to reasons previously reported on—domestic-worker employers being workers themselves and therefore not at home at the time when inspections are routinely done, or domestic workers’ place of work being a private household and the employer therefore having the right to deny entry. The effort and resources required to follow up with employers of only one individual would require many inspectors that could be more efficiently deployed to businesses employing many staff [15]. Furthermore, it is unclear what the consequences for employers of non-compliance are. Legislation provides for penalties through fines or imprisonment [43], but it is unknown whether any domestic-worker employers have faced such penalties for non-compliance. Other suggestions for compliance have been that the government should work with technology partners to improve the speed and ease of compliance and that incentives be provided for compliance (such as those related to taxes) [44]. There is also the potential for digital solutions, such as the development of a mobile application, to assist with improving access to the current social insurance programme, including for non-standard workers.

#### 4.2.3. Unclear Guidance on Certain Components of Maternity Protection

Health care in SA is available free of charge to women during pregnancy and the postnatal period, but access to these benefits requires the ability to attend clinic visits. Public health systems, however, are mostly only open during working hours, do not make specific appointment times, and usually have long waiting times before patients are attended to. Furthermore, these facilities are mostly nearby women’s houses, which are often far from their place of work, creating further logistical challenges in accessing healthcare during the workday. These challenges are not unique to domestic workers and are similar for women employed formally, although domestic workers and other informal workers are disproportionately affected due to geospatial inequalities in SA. In this research, domestic workers reported being able to access medical benefits (antenatal and postnatal clinic check-ups) by organising to attend the clinic on their days off work, effectively using unpaid leave for this purpose. In SA law, the *Code of Good Practice on the Protection of Employees During Pregnancy and After the Birth of a Child* recommends that arrangements be made for pregnant and breastfeeding employees to attend antenatal and postnatal clinics [45]. However, this is a recommendation in a Code of Good Practice and therefore not legally enforceable [14], and no specific guidance is provided on how these arrangements should be made (i.e., how many days or hours women should be entitled to and what type of leave it should be categorised as). This means that implementation of this recommendation is inconsistent and employer dependent, and therefore, as seen in the results of this research, often the responsibility of the woman to organise in her own time.

#### 4.2.4. Breastfeeding Breaks and Childcare Components of Maternity Protection Should Be Accessible to Domestic Workers

Certain components of maternity protection should be amenable to being made available to domestic workers, such as the provision of breaks to breastfeed or express milk. Although one participant described potential difficulty in breastfeeding her child at work due to physical distance, it should be feasible for domestic workers to be able to express their milk while at work. Since domestic workers’ place of work is a private household, it should be simple to guarantee privacy. Although some participants in this research thought there may be challenges to storing expressed breastmilk in the employer’s fridge, this is something that would probably be employer dependent. Alternatively, a domestic worker could bring a small cooler box with her to work in which to store expressed breastmilk. Research from the United States in a formal work setting has shown stigma associated with women pumping breastmilk at work [46]. That many participants in our study were not even aware of breaks to breastfeed or express means that this is a component of maternity protection that has potential for improved implementation. Advocacy, awareness, and education are required on the recommendation for breastfeeding breaks in the Code of Good Practice. It would be helpful for employers and workers to be made aware of the benefits that breastfeeding could not only contribute to the health and financial situation of domestic workers and their infants, but also potential employer benefits such as worker productivity and reduced absenteeism. This is an example of small accommodations that could contribute to improved early-life nutrition. Furthermore, advocacy is required to destigmatise breastfeeding breaks and breastmilk expression at work.

Domestic workers in this research described that childcare is expensive and access to childcare problematic. Childcare is not a component of the ILO Maternity Protection Convention, but childcare for infants and children up to three years of age is described in the ILO Maternity Protection Resource Package [2,47]. The ILO recommends that affordable, appropriate care services for children, whether in the child’s home, in a childminder’s home, or centre-based care, should be available as services that lighten the load of unpaid care work for working mothers. It has previously been recommended that access to quality, accessible, public childcare services is a key policy intervention that could improve the productivity and income of women working informally [48]. Childcare for domestic workers is also a component of maternity protection that can be employer dependent, since some participants in this research reported being allowed to bring their child with them to work and others indicated that it would not be deemed appropriate by the employer.

### 4.3. International Accountability for Maternity Protection in SA Should Be Ensured

SA has not ratified the ILO Maternity Protection Convention [49]. Ratification of the convention could provide greater pressure on government to ensure that comprehensive maternity protection is available and accessible to all workers.

### 4.4. Limitations

Since participants were recruited from a survey distributed from an online platform employing domestic workers, most participants were employed through the platform. The sample was not nationally representative. Less than 1% of domestic workers in SA are active on the platform [10]. Certain responses therefore reflect unique characteristics of this employment arrangement. The recruitment process also prescribed that respondents needed to be digitally literate and have access to a smartphone and internet (data or Wi-Fi), therefore biasing the sample. Neither live-in domestic workers nor foreign or migrant domestic workers were included in the sample of women interviewed during IDIs, and these are both vulnerable groups and make up a large segment of the domestic-worker population. Only domestic workers in Cape Town were included in the individual IDIs, and therefore only one urban-city and no rural domestic workers were included. Despite efforts to address researcher bias and reflexivity, it is possible that some bias remains in the interpretation of the data.

## 5. Conclusions

The results of this research show that domestic workers in the Western Cape are currently unable to access most components of comprehensive maternity protection. Many domestic workers are unaware of their rights and employers are ill-informed of their obligations. Improved access to all components of maternity protection would result in improved quality of care for women during pregnancy, around the time of childbirth and on return to work, and for their newborns, especially if an enabling environment for breastfeeding were created. Comprehensive maternity protection for all domestic workers could contribute to improved care with subsequent health and development benefits for a vulnerable group of working women and their children.

## Figures and Tables

**Figure 1 ijerph-20-02796-f001:**
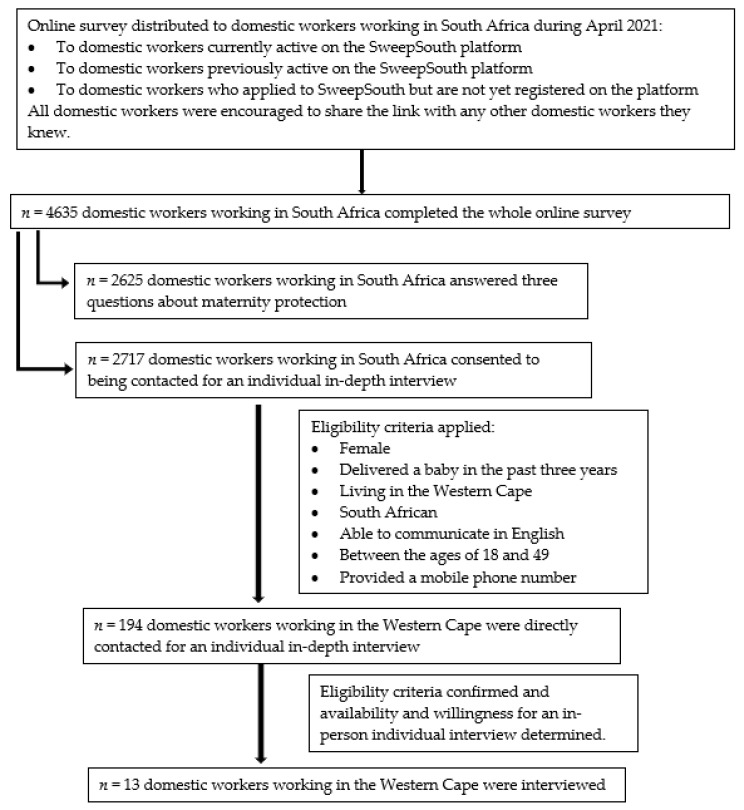
Flow chart illustrating the participant sampling and selection process.

**Table 1 ijerph-20-02796-t001:** Summary of maternity-protection entitlements for working women in SA.

Component of Maternity Protection	Provisions in South African Policy and Legislation
**Maternity leave**	All workers entitled to four consecutive calendar months of unpaid maternity leave in accordance with the *Basic Conditions of Employment Act*.
**Cash benefits**	Women working at least 24 h per month are entitled to social insurance, whereby employers and employees make monthly contributions to the Unemployment Insurance Fund (UIF) and women can claim two-thirds of their earnings (up to a maximum threshold) as maternity benefits. This is mandated by the *Unemployment Insurance Act*.
**Medical benefits**	In SA, access to healthcare is available to all, including pregnant and breastfeeding women, through public healthcare services guaranteed by the *Constitution*.
**Health protection**	A *Code of Good Practice on the Protection of Employees During Pregnancy and After the Birth of a Child* contains guidance on health protection for pregnant and breastfeeding women.
**Employment protection**	All pregnant women in SA are entitled to job security, since dismissal related to pregnancy is prohibited by the *Labour Relations Act*.
**Non-discrimination**	All women in SA are protected by non-discrimination due to pregnancy through the Constitution.
**Breastfeeding breaks**	The *Code of Good Practice on the Protection of Employees During Pregnancy and After the Birth of a Child* recommends twice daily breastfeeding breaks of 30 min for all working women until their child is six months old, but this is not legislated.
**Childcare support**	There is no legislation on childcare support for working women in SA.

Source: [15] Pereira-Kotze, C.; Malherbe, K.; Faber, M.; Doherty, T.; Cooper, D. Legislation and Policies for the Right to Maternity Protection in South Africa: A Fragmented State of Affairs. *J. Hum. Lact.*
**2022**, *38*, 686–699. https://doi.org/10.1177/08903344221108090 (accessed on 12 January 2023).

**Table 2 ijerph-20-02796-t002:** Socio-demographic characteristics of domestic workers who completed the survey *(N* = 2625).

Characteristic		n	%
**Sex**	Female	2537	96.7
	Male	82	3.1
	Other/prefer not to say	6	0.2
**Age**	Average: 35.5 years; range of 19–62 years		
**Nationality**	Zimbabwe	1514	57.7
	South Africa	1031	39.3
	Malawi	32	1.2
	Lesotho	21	0.8
	Democratic Republic of the Congo	20	0.8
	Other ^a^	7	0.3
**Province of work**	Gauteng	1453	55.4
	Western Cape	1006	38.3
	KwaZulu-Natal	148	5.6
	Other ^b^	18	0.7
**Number of children**	None	99	3.8
	One	553	21.1
	Two	1022	38.9
	Three	659	25.1
	Four	224	8.5
	Five	53	2
	Six or more	15	0.6
**Earnings from domestic work (per month)**	Less than R1500 ^c^	542	20.7
	ZAR 1501–2000	371	14.1
	ZAR 2001–3000	828	31.5
	ZAR 3001–4000	536	20.4
	ZAR 4001–5000	239	9.1
	ZAR 5001–6000	76	2.9
	ZAR 6001–7000	26	1
	More than ZAR 7000	7	0.3
**Registered for the UIF**	Yes	203	7.7
	No	2124	80.9
	Do not know	298	11.4

^a^ Other (*n* = 7): Namibia 2, Mozambique 2, Cameroon 1, Nigeria 1, Rwanda 1; ^b^ Other (*n* = 18): Eastern Cape 7, Mpumalanga 4, Limpopo 3, North-West 2, Free State 1, Northern Cape 1; ^c^ USD 1 = ZAR 17.72, therefore ZAR 1500 = USD 84.62 and ZAR 7000 = USD 394.91 (7 November 2022) [26]. UIF, Unemployment Insurance Fund.

**Table 3 ijerph-20-02796-t003:** Domestic workers’ perceptions of maternity-protection entitlements (*N* = 2625).

Do you think that a domestic worker who is pregnant at the moment is allowed to receive any of the following benefits? (Choose all that apply.)	Yes	%
Paid time off work during her pregnancy to attend pregnancy (antenatal) clinic visits.	1762	67.1
Unpaid time off work during her pregnancy to attend pregnancy (antenatal) clinic visits.	149	5.7
Have her employer make changes to the tasks she has to carry out during her work so as not to cause any harm to her or her baby during her pregnancy (for example, not having to lift heavy objects or bend over towards the end of her pregnancy).	1181	45.0
She should not be allowed to lose her job just because she is pregnant or will be having a baby.	1359	51.8
She should not be discriminated against because she is pregnant or will be having a baby (for example, her pay should not be reduced because she is pregnant; if starting with a new employer, the employer should not state that she cannot fall pregnant).	1224	46.6
Do not know.	140	5.3
**If you, or a domestic worker in a similar position to you, were to fall pregnant and have a baby, which maternity benefits do you think you or she would be able to receive? (Select only ONE option.)**		
No maternity leave, or less than 6 weeks leave (after the baby is born).	100	3.8
Some maternity leave (more than 6 weeks and less than 4 months of leave after the baby is born).	546	20.8
Four months of unpaid maternity leave.	52	2.0
Four months of partially paid maternity leave.	525	20.0
Four months of maternity leave and can claim from the UIF.	563	21.5
Four months of full paid maternity leave (organised by the employer).	687	26.2
Do not know what is allowed.	152	5.8
**Do you think that when a domestic worker returns to work after maternity leave she is allowed to: (Choose all that apply.)**		
Take paid time off work to attend baby (postnatal) clinic visits?	1579	60.2
Take unpaid time off work to attend baby (postnatal) clinic visits?	411	15.7
Take daily breastfeeding breaks (at least one break during the working day to either express breastmilk or breastfeed the baby)?	479	18.3
Bring her baby to work with her?	177	6.7
None of the above.	278	10.6
Do not know.	245	9.3

**Table 4 ijerph-20-02796-t004:** Themes and subthemes from the IDIs.

Components of Maternity Protection	Sub-themes
**Health protection at the workplace**	● Health protection at the workplace for domestic workers is employer-dependent
**Medical benefits while on maternity leave**	● Access to health care for domestic workers during the antenatal and postnatal periods necessitates unpaid leave
**Non-discrimination and job security for pregnant and breastfeeding women**	● Some domestic workers experience discrimination due to pregnancy and childbirth ● Many domestic workers experience job insecurity due to pregnancy and childbirth
**Cash benefits while on maternity leave**	● Domestic workers are unable to access cash payments while on maternity leave despite legal eligibility to social insurance in SA ● Inaccessibility to social insurance results in dependence on social assistance, which also has challenges
**A period of maternity leave**	● Unpaid maternity leave is available to domestic workers but is unaffordable and therefore inaccessible
**Breastfeeding (or expressing) breaks on return to work**	● Domestic workers are not familiar with the entitlement to breastfeeding breaks upon return to work ● Challenges with storing expressed breastmilk at work
**Support to access childcare**	● Domestic workers struggle to access childcare on return to work

## Data Availability

The datasets generated and/or analysed during the current study are not publicly available to protect participant confidentiality but are available in an anonymised form from the corresponding author on reasonable request.

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
