# Peer review of "Access to Maternity Protection and Potential Implications for Breastfeeding Practices of Domestic Workers in the Western Cape of South Africa"

_ijerph, 2023, doi:10.3390/ijerph20042796_

Round 1

Reviewer 1 Report

This research addresses an important topic and takes an excellent approach to the study.

The introduction needs to have a section which details who is eligible for what benefits in South Africa (more than just noting the system is fragmented) in order to understand the survey results in context.

The methods need to address why so few people received individual interviews.

Results:  The individual interviews should be used to illuminate points in the survey results and generate hypotheses for future research rather than as substantial freestanding results given that only 13 interviews were carried out.

Discussion: needs to more clearly distinguish between whether the results suggest domestic workers had benefits but were unaware of them, did not receive benefits because of employers or because they had not paid into the system etc in order to get to the implications.

Reviewer 2 Report

This manuscript presents a mixed-method cross-sectional study on access to comprehensive maternity protection and the implications for breastfeeding practices among South African domestic workers in the Western Cape using data from a quantitative online survey of domestic workers employed through an online platform and individual in-dept interviews conducted in Cape Town.

The manuscript is well-structured and articulated and is covering a topic requiring much more attention from research to evidence-based policy formulation and enforcement. As we celebrate over a century of the maternity protection convention and subsequent updates, it is obvious that this has been a neglected field now to be illuminated both globally and in South Africa. 

Although the sample is not nationally representative for South Africa or the diversity of the informal workerforce, it is a useful contribution to the literature and evidence-base for the country, regionally, and globally using the example of domestic workers recruited via an online portal. The methodology is therefore novel and tapping into the opportunities of the massive digital transformation. 

The manuscript is based on my comprehensive assessment ready for acceptance in the current format, but I have provided some detailed feedback for the authors considerations and to be potentially addressed during the final proofreading.

Title: Consider replacing “domestic workers in South Africa” with “national domestic workers in the Western Cape of South Africa” since this is not a nationally representative sample.

Page 1, line 40-41: Consider also including the 2030 WHA target for exclusive breastfeeding to align with the SDGs.

Page 1, line 44 and throughout the paper: Consider using “protection” as the plural form instead of “protections” to align with other literature on the topic (ref: Global Breastfeeding Collective led by UNICEF and WHO).

Consider replacing “breast milk” with “breastmilk” in one word throughout the paper to also align with updated recommended terminology from UNICEF and WHO.

Page 3, line 138: Consider referring to CPK as the first author for better understanding and then the abbreviation in brackets. I thought at first it was a technology or tool referred to for conducting the interviews. Alternatively, say “one researcher” and then the abbreviated name in brackets as done later in the paper.

Results / table 1: Do you mean by gender or by sex?

An important message in the text could come out clearer in the abstract and / or conclusion and also some related proposed solution: Domestic workers need to know their rights and individual employers needs to know their obligations to fulfil them. Could social and behaviour change communication campaigns be considered a solution to address this? Or for this group it might work with a better digital solution to access the needed information since they are already recruited through an online portal. Consider elaborating more on practical and policy solutions based on the data and literature.

In the discussion, consider also elaborating on the difference between social insurance and social assistance schemes as a solution for informal workers including any guidance from ILO or other normative agencies. My understanding and experience from country policy advocacy is that ILO would primarily recommend social assistance over social insurance (volunteer contribution scheme) for informally employed workers, but the context might be different in South Africa.

This discussion could potentially also elaborate further on the finding related to complicated and time-consuming procedures to access cash payments (related to my comment on potential for digital solutions in line with the country’s digital transformation agenda, if available).

Congratulations to the co-authors or this excellent manuscript and thank for considering some of my suggestions above in the finalization of this publication.

Reviewer 3 Report

Please find my suggestions on the attached PDF file. 

Reviewer 4 Report

All of my suggestions for the authors are in the attached file. 

Round 2

Reviewer 3 Report

I would like to thank the authors for their extensive editing work. 

Here are a few more minor comments: 

Lines 189-190: As the authors used the six components of comprehensive maternity protection as a framework for analysis, a priori the themes should correspond with these components. However, as the authors became familiar with the content of the interviews, they may have identified and developed sub-themes. Thus, I invite the authors to modify the following sentence, as the initial themes are not generated “Next, initial themes were generated, which were reviewed, developed, and refined”.

Line 382: I suggest modifying the sub-headings as follows “Difficulties for domestic workers to access paid maternity leave”

Lines 561-565: I propose to modify the sentence as follows: “From our results, it seems that the inaccessibility of maternity protection for domestic workers is due to a variety of reasons, including limited knowledge and awareness of both domestic workers and their employers regarding entitlements, as well as systemic problems with the implementation of these provisions entitlements.”

Line 645: I suggest modifying the sub-headings as follows “4.2.1 There are Challenges in the implementation of social insurance in SA”.

Line 672: I suggest modifying the sub-headings as follows “4.2.2 There are Limitations to the enforcement of maternity protection legislation in SA”

Line 675: It seems that a spelling error has occurred “Inspections at households may be complex to organise17rganize, due to reasons previously reported on”

Line 691: I suggest modifying the sub-headings as follows “4.2.3 There is Unclear guidance on certain components of maternity protection”
